# Slicing Ceramics on Material Removed by a Single Abrasive Particle

**DOI:** 10.3390/ma13194324

**Published:** 2020-09-28

**Authors:** Yao-Yang Tsai, Ming-Chang Wu, Yunn-Shiuan Liao, Chung-Chen Tsao, Chun-Yao Hsu

**Affiliations:** 1Department of Mechanical Engineering, National Taiwan University, Taipei 10617, Taiwan; yytsai@ntu.edu.tw (Y.-Y.T.); liaoys@ntu.edu.tw (Y.-S.L.); 2Department of Mechanical Engineering, Lunghwa University of Science and Technology, Taoyuan 33306, Taiwan; AA021@mail.lhu.edu.tw

**Keywords:** wire-saw machining, material removal processing, brittle fracture, plastic deformation

## Abstract

Multi-wire saw machining (MWSM) used for slicing hard-brittle materials in the semiconductor and photovoltaic industries is an important and efficient material removal process that uses free abrasives. The cutting model of single-wire saw machining (SWSM) is the basis of MWSM. The material removal mechanism of SWSM is more easily understood than MWSM. A mathematical model (includes brittle fracture and plastic deformation) is presented in this paper for SWSM ceramic with abrasives. This paper determines the effect of various machining parameters on the removal of hard-brittle materials. For brittle fracture of SWSM ceramics, the minimum strain energy density is used as a fracture criterion. For plastic deformation of SWSM ceramics, the material removal is calculated using equations of motion. Actual wire-sawing experiments are conducted to verify the results of the developed mathematical model. The theoretical results agree with experimental data and practical experience. From the developed mathematical model, brittle fracture plays a major role in material removal of SWSM ceramics. Wire speed (S) and working load (P) are positively correlated with material removal of SWSM ceramics. The coefficient of friction is low, a lateral crack, which propagates almost parallel to the working surface, leads to more brittle fracture and material removal is increased.

## 1. Introduction

Ceramics are an important hard-brittle material for the semiconductor and photovoltaic industries. Wire-sawing technology (contains SWSM and MWSM) is the most efficient and economic tool for machining hard-brittle materials, such as silicon, SiC, ceramic, and sapphire for the photovoltaic or semiconductor industries [1,2,3]. A single-wire saw consists of a thin steel wire wound around wire-guides, forming a web of parallel wires (namely SWSM). The abrasion for single-wire sawing technology can be free or fixed. Wire-saw machining (WSM) is a very important part of slicing ceramic ingots. The efficiency of SWSM primarily depends on the cutting action of abrasive particles in the slurry. The ceramic removed by the abrasive particles is the major factor in machining efficiency, cost reduction, and the quality of the machined surface [4]. Studies [5,6] show there are both cracks and plastic flow during the cutting of brittle materials. Finnie and McFadden [7] used hard abrasive particles to strike a ductile metal at a low angle and predicted the volume of material removed is positively correlated to the nth power of speed (for 2 < n < 3). Evans [8] showed the impact of each abrasive particle on hard-brittle materials is proportional to the penetration depth of the abrasive particle and the square of the crack length using the elastic-plastic theory. Wang and Rajurkar [9] studied the cutting factors on the material removal rate (MRR) of an ultrasonic machining process. However, the MRR is the amount of weight removed per unit volume of material and is widely used to optimize the wire-sawing process. Slicing ceramics requires a high value for the MRR, but the processing conditions must be controlled to avoid instability, which causes defects such as cracks, porosity, and straying of the cutting path. Some recent research on a WSM stone cutting system is also presented [10,11].

On the other hand, some studies developed a few material removal models for the wire-sawing process using abrasive diamond particles [12,13,14,15]. The removal of the hard-brittle material during wire-sawing involves plastic deformation and brittle fracture. During SWSM, the stainless steel wire driven abrasives move against rather than strike the work material. Therefore, the present study determines the theoretical model of MRR for hard-brittle material during SWSM. The governing equations are derived based on the fracture principle of strain energy density (for brittle fracture) and the equation of motion (for plastic deformation), and the effects of various cutting parameters (S and P) on material removal are determined. In addition, each experiment was replicated three times to verify the theoretical model developed in this study.

## 2. Theoretical Derivation Model

### 2.1. Removal Mechanism of Hard-Brittle Material

The removal mechanism for the hard-brittle material during WSM involves the shear force of the thin film in the slurry and the fine cutting from the rolling and impact of abrasives [16]. The shear force of the thin film and the rolling impact have a limited effect due to large abrasives of about 8–27 µm being used for WSM. When the cut depth is less than 1 µm, the hard-brittle materials experience ductile-regime grinding [17]. The material experiences plastic deformation due to the plowing of abrasives and piles up either side of the abrasive. Material at the front end of the abrasive cutting edge also undergoes chipping due to plastic deformation. For a cut depth of 1–4 µm, a lateral crack emerges along with debris [18]. This result is similar to that for scribing indentation [19], in which the pile-up generated by plowing function may be removed at the same time.

Figure 1a,b illustrates the removal mechanism for the hard-brittle materials during SWSM. Figure 1a shows the material is pushed to the sides and piles up due to plowing by abrasives when the cut depth is shallow, and there is fine chipping at the front end of the abrasives. Figure 1b shows the lateral crack formed for a deeper cut, which produces debris and removes a larger quantity of material. The experimental results for driving stainless steel wire to allow SiC slurry to work on a workpiece made of Al_2_O_3_ are shown in Figure 2a,b. Position A in Figure 2a shows the plastic deformation and position B display brittle fracture. In Figure 2b, position D denotes the chipping generated by brittle fracture. The chip was analyzed using an Energy-dispersive Spectrometer (EDS). Figure 2c shows the chip consists of Al_2_O_3_, in which Au is tinted for observation by SEM. This study assumes a minute volume is removed by abrasive slurry, as shown in Figure 3, wherein zones I and II are the plastic deformation regions, and zone III is the lateral fracture region, respectively. The following descriptions are divided into two parts.

### 2.2. Brittle Fracture

The stress intensity factor (K), the energy release rate (G) and the J-integral are the parameters commonly used to predict fracture within a brittle material. However, they all assume there is a crack before the analysis. Sih [20] uses the strain energy density criterion to predict fracture initiation and to determine the fracture trajectory. The strain energy density criterion is based on the global energy field, instead of the local stress or energy field for the failure analysis. This method does not require the assumption a crack exists. The strain energy density (dW/dV) can be written as follows:(1)$$\frac{dW}{dV}=\int_{0}^{\varepsilon_{ij}}\sigma_{ij}d\varepsilon_{ij}$$
where Wdenotes the total energy stored in an element; Vdenotes the volume of material removed by an abrasive particle and $\sigma_{ij}$ and $\varepsilon_{ij}$ are the stress and strain vectors, respectively. Using the strain energy density, the location and the instant of fracture initiation require the following basic assumptions [19]:

(1). Initial fracture occurs at the location with the minimum value of (dW/dV), denoted by ${\left(dW/dV\right)}_{\min}$. If there is more than ${\left(dW/dV\right)}_{\min}$, then fracture occurs at the point between the minimum value of ${\left(dW/dV\right)}_{\min}$ and maximum value of (dW/dV)_max_, denoted by ${\left(dW/dV\right)}_{\min}^{\max}$.

(2). Fracture extension occurs when the (dW/dV) reaches a critical value, denoted by ${\left[\left(dW/dV\right)\right]}_{C}$.

The local view of the strain energy density of the crack tip uses the polar coordinates, *r* and *θ*. The fracture path is predicted based on assumption (1). There is a ${\left(dW/dV\right)}_{\min}^{\max}$ around the specific point for a given *r* and *θ* between −90° and 90°. Point A in Figure 4 is the location where the fracture occurs. As *r* increases gradually, a series of points corresponding to ${\left(dW/dV\right)}_{\min}^{\max}$ are obtained. The fracture path is obtained by connecting these points [21]. There are two different types of cracks: a median crack that propagates underneath scribing and a lateral crack, resulting in surface fracture. The lateral crack has a significant effect on the removal of brittle material. The minimum threshold load for a lateral crack (*P*^*^) is derived in [18]. In Figure 3d, the concentrated forces of the abrasive particle acting in the *Z* and *−Y* directions on the sides of the workpiece (*Z* plane) are $P^{*}\cos\alpha (P_{ZZ})$ and $P^{*}\sin\alpha (-P_{ZY})$, respectively, so the stress in the *Z* plane, expressed in polar coordinates, is [22]:(2)$$\sigma_{r}=\frac{2P^{*}}{\pi r}\cos{\left(\alpha +\theta \right)}^{}$$
where tanα=μ, and μ denotes the coefficient of friction. The strain energy density is [22]
(3)$$\frac{dW}{dV}=\frac{\left(1-2\nu^{2}\right)}{2E}\sigma_{r}^{2}$$
where ν is Poisson’s ratio and E is the modulus of elasticity. Substituting Equation (2) into Equation (3) yields:(4)$$\frac{dW}{dV}=\frac{\left(1-2\nu^{2}\right)}{2E}{\left(\frac{2P^{*}}{\pi_{}r}\right)}^{2}\cos^{2}(\alpha +\theta )$$

According to the uni-axial tensile test, the critical strain energy density ${\left(dW/dV\right)}_{c}$ can be represented as:(5)$${\left(\frac{dW}{dV}\right)}_{C}=\frac{\sigma_{U}{}^{2}}{2E}$$
where $\sigma_{U}$ is the ultimate material strength. When ${\left(dW/dV\right)}_{\min}^{\max}={\left(dW/dV\right)}_{c}$, the material begins to fracture, so:(6)$$\frac{\left(1-2\nu^{2}\right)}{2E}{\left(\frac{2P^{*}}{\pi_{}r}\right)}^{2}\cos^{2}(\alpha +\theta_{c}^{})=\frac{\sigma_{U}{}^{2}}{2E}$$
where $\theta_{c}$ is the angle at which ${\left[dW/dV\right]}_{\min}^{\max}$ occurs.

Using Equation (6), for a specific radius r and if θ is −90° to 90°, the location corresponding to ${\left(dW/dV\right)}_{\min}^{\max}$ can be determined. For coefficients of friction (μ) of 0.1, 0.2, 0.3, and 0.4, the locations for fracture initiation are derived using Equation (6). The calculated values of $\theta_{c}$ are respectively equal to −5.71°, −11.3°, −16.7°, and −21.8°.

### 2.3. Plastic Deformation

Figure 5a shows a mass (m) of abrasive particle plowing a trajectory in a ductile material because of its inertial force. Finnie and Mcfadden [7] solved the equations of motion for this particle to predict the volume removed by erosion. The equations of motion for abrasive particles are used to derive the volume of plastic deformation, as shown in Equations (7)–(9). In Figure 5b, the center of mass of the abrasive particle moves parallel to the *Z* plane and rotates at an angle of ϕ due to the working load (P) that is exerted by the stainless wire and the wire speed (S). The locus left by the abrasive particle tip cutting into the material surface is (*X_T_*, *Y_T_*). The cutting also leads to a plastic deformation chip at the front end:(7)$$-F_{X}=mX^{"}$$
(8)$$-F_{Y}+P=mY^{"}$$
(9)$$F_{X}r=I\phi^{"}$$

The cut depth is very shallow and the rotational angle is very small, so the movement relative to the abrasive particle tip is assumed and the center of mass is $X_{T}≅X+r\phi$ and $Y_{T}=Y$, as shown in Figure 5b. The horizontal and vertical cutting forces at the tip of the abrasive particle are $F_{X}$ and $F_{Y}$, respectively, as shown in Figure 5b, so the plastic flow stress $\sigma =F_{X}/A$ is assumed to be constant, where A is the projection area of the contact between the abrasive particle and the chip. However, A=Lb, where *L* is the chip length and *b* is the width of cut for a single abrasive particle, as shown in Figure 5b. From reference [7], $L/Y_{T}=K=2$, $F_{Y}/F_{X}=J$, where *J* is a constant.

Using these assumptions, $F_{X}=\sigma A=\sigma Lb=\sigma bKY_{T}=\sigma bKY$. Similarly,$F_{Y}=\sigma bKJY$ and the centroid moment $M_{G}=\sigma bKrY$. The equations of motion for the abrasive particle are rewritten as:(10)$$mX^{"}+\sigma bKY=0$$
(11)$$mY^{"}+\sigma bKJY=P$$
(12)$$I\phi^{"}-\sigma bKrY=0$$
where *m* is the mass for a single abrasive particle, as shown in Figure 3a, *P* is the working load, *I* is the moment of inertia of a particle about its center of gravity, and *r* is the average particle radius, as shown in Figure 5b. For the initial conditions, Y(0)=0 and $Y^{\prime }(0)=0$, the solution of Equation (11) is:(13)$$Y(t)=\left({-}_{}\frac{{}_{}P}{K_{}b_{}\sigma_{}J^{}}\right)\cos_{}{}_{}{\left(\sqrt{\frac{K_{}b_{}\sigma_{}J}{m}}\right)}_{}t+\frac{P}{K\;b\;\sigma \;J}$$

By substituting Y(t) into Equation (10) and using initial conditions X(0)=0 and $X^{\prime }(0)=S$, then X(t) can be expressed as:(14)$$X(t)={\left({-}^{}\frac{{}_{}P}{K_{}b_{}{}_{}\sigma_{}J_{}{}^{2}}\right)}_{}cos_{}{}_{}{\left(\sqrt{\frac{K_{}b_{}\sigma_{}J}{m}}\right)}_{}t_{}{-}_{}{\frac{P}{2mJ}}_{}t_{}{}^{2}+S_{}t{+}_{}\frac{P}{K_{}b_{}\sigma_{}J_{}{}^{2}}$$

By substituting Y(t) into Equation (12) and using initial values ϕ(0)=0 and $\phi^{\prime }(0)=0$, the rotation of the particle is:(15)$$\varphi (t)={\left(\frac{mrP}{K_{}b_{}{}^{}\sigma IJ_{}{}^{2}}\right)}_{}cos_{}{}_{}{\left(\sqrt{\frac{K_{}b_{}\sigma_{}J}{m}}\right)}_{}t{+}^{}{\frac{rP}{2mJ}}_{}t_{}{}^{2}{-}^{}\frac{mrP}{K_{}b_{}\sigma_{}I_{}^{}J_{}{}^{2}}$$

### 2.4. Material Removed by an Abrasive Particle

The material removed by an abrasive particle (V) is calculated by combining the brittle fracture and the plastic deformation as:(16)$$V=\int b^{*}Y_{T}dX_{T}$$
where $X_{T}=X(t)+\gamma \phi (t)$
$$Y_{T}=Y_{\left(t\right)}$$
(17)$$b^{*}=b+2Y_{T}(t)\cot\alpha$$

Therefore, the material that is removed is:(18)$$V=\int \left[b+2Y(t)\cot\alpha ]Y_{T}d[X(t)+\gamma \phi (t)\right]$$

If the tip of the abrasive particle is moving horizontally as it leaves the surface, $Y_{T}(t)=0$*,* so Equation (10) yields:(19)$$\cos{\left(Kb\sigma J/m\right)}^{1/2}t=1\;\mathrm{or}\;t=2\pi /{\left(Kb\sigma J/m\right)}^{1/2}$$

Substituting Equations (13)–(15), and (19) into Equation (18), the volume that is removed is written as:$$V={\left(\frac{mr^{2}{}_{}P}{IK_{}b_{}\sigma_{}J^{2}}{-}_{}\frac{P}{Kb_{}\sigma_{}J^{2}}\right)}_{}\left[\frac{3_{}P}{4K\sigma_{}J}+\frac{5}{3}{\left(\frac{P}{Kb_{}\sigma_{}J}\right)}^{2}cot_{}\alpha \right]\;+{\left(\frac{P}{mJ}{-}_{}\frac{r^{2}{}_{}P}{I_{}J}\right)}_{}\left[\frac{P}{K\sigma_{}J}+\frac{15}{4}{\left(\frac{P}{Kb\sigma_{}J}\right)}^{2}cot_{}\alpha \right]\frac{m}{Kb_{}\sigma_{}J}\;+[{\left(\frac{r^{2}{}_{}P}{I_{}J}-\frac{P}{m_{}J}\right)}^{}{\left(\frac{P}{Kb\alpha_{}J}\right)}^{2}cot\alpha +\left(\frac{P}{K\sigma_{}J}+3_{}{\left(\frac{P}{Kb\sigma_{}J}\right)}^{2}cot\alpha \right)]\frac{2\pi S}{\sqrt{Kb_{}\sigma_{}J/m}}\;+{\left({\frac{r^{2}{}_{}P}{I_{}J}}_{}{-}_{}\frac{P}{mJ}\right)}_{}^{}{\left[\frac{P}{2K\alpha_{}J}+\frac{1}{2}{\left(\frac{P}{K_{}b\alpha_{}J}\right)}^{2}cot_{}\alpha \right]}_{}^{}{\left(\frac{2\alpha }{\sqrt{Kb_{}\alpha_{}J/m}}\right)}^{2}$$

$I≅mr^{2}$, so the material that is removed is written more simply as follows:$$V=\left[\frac{P}{K\sigma J}+3{\left(\frac{P}{Kb_{}\sigma_{}J}\right)}^{2}cot\alpha \right]\frac{2\pi S}{\sqrt{Kb_{}\sigma_{}Jm}}$$$\sigma =F_{X}/A$, $L/Y_{T}=K=2$, tanα=μ and $F_{Y}/F_{X}=J$, so:(20)$$V=\left[\left(\frac{A_{}P}{2F_{Y}}\right)+{\frac{3}{\mu }}_{}{\left(\frac{A_{}P}{2bF_{Y}}\right)}^{2}\right]\frac{2\pi S}{\sqrt{2bF_{Y}/mA}}$$

This equation consists of two parts: for a plastic deformation and for a brittle fracture. The material removal increases as the wire speed (*S*) and working load (*P*).

## 3. Experimental Results and Discussion

Experiments were conducted to test the proposed theory. The workpiece swings back and forth, as shown in Figure 6. This vibrating machining model evenly distributes the cutting force of each active grain and makes the disposal of chips more efficient. The main experimental parameters and their values are listed in Table 1. To derive a more precise correlation between the machining parameters and the material that is removed, Equation (20) is expressed in non-dimensional form as:(21)(V2bFY/mA2πS)(AP2FY)=1+3μ(AP2FY)1b2

Let $N_{X}=1+\frac{3}{\mu }\left(\frac{AP}{2F_{Y}}\right)\frac{1}{b^{2}}$ and NY=(V2bFY/mA2πS)(AP2FY).

For a single abrasive particle with GC# 600 (Green Silicon Carbide No. 600 mesh), the width of cut *b* (D/3) is about 9 µm, where D is the diameter, which is 27 µm, and the mass *m* ($\pi \rho D^{3}/6$) is 2 × 10^−11^ kg, where ρ is the density, with a value of 3.22 g/cm^3^. If µ = 0.3 [23], $F_{Y}/F_{X}=J≅1$ [7], the plastic flow stress σ for Al_2_O_3_ is about 400 MPa, chip length is *L* (1.1*b*) and the horizontal cutting force $F_{X}$ is (σbL=0.0356N). For a single abrasive particle, the working load is *P* (total working load/$n_{0}$), so, if $a_{0}$ is the total area of the machining zone (See Figure 6), the maximum number of particles participating in the operation is calculated by dividing the area of the grit by the largest cross-section a_0_ ($L_{0}b_{0}$), where $L_{0}$ is the line contact length and $b_{0}$ is the line width. Accounting for tiny gaps between the grit, the total number of particles $n_{0}$ is: $n_{0}=C_{0}a_{0}/(\pi D^{2}/4)$, where $C_{0}$ is the gap coefficient, which has a value of 10–20%.

The material V removed by a single abrasive particle is $V_{0}/n_{1}$, where $V_{0}=A_{0}w$ denotes the total volume removed per unit of time, where $A_{0}$ is the cutting area per unit of time and w is the kerf width. $n_{1}(C_{1}Sn_{0}/L_{0})$ denotes the total number of machining abrasives per unit of time, where $C_{1}$ is the wire speed coefficient, which has a value of 70–90%, and *S* is the wire speed. The experimental conditions and results are listed in Table 2, which is used to produce the chart in Figure 7. The figure shows the experimental values with different wire speeds (4.1 m/s (●), 5.6 m/s (■), or 6.4 m/s (▲)); all have an approximate linear relationship, which is very close to the format for theoretical Equation (18). However, if the working load is increased, the clearance between the wire and the surface of the workpiece decreases, which prevents the abrasive grains from entering the working area, so the experimental material removal rate is lower than the theoretical value.

The $N_{Y}$ value for 5.6 m/s (■) is lower than 4.1 m/s (●) because the volume that is removed by one single abrasive is $V=V_{0}/n_{1}$, where $V_{0}$ denotes the total volume removed per unit time, n_1_ denotes the total number of machining abrasives per unit time and $n_{1}=C_{1}Sn_{0}/L_{0}$. The speed of the wire-saw is unknown, so the ratio of the speed to the total number of machining abrasives per unit of time ($n_{1}$) is assumed to be 1:1. Therefore, the faster the wire, the greater the value of n_1_, and the smaller the volume removed by a single abrasive (*V*). The 5.6 m/s (■) value at a high wire speed is lower than the 4.1 m/s (●) value at a low wire speed. The result verifies the equation for the material removed by a single abrasive Equation (20).

Equation (20) allows the contribution of the material removed by plastic deformation to be determined. This is plotted as a dashed line in Figure 8. A small amount of material is removed by plastic deformation, and brittle fracture plays the dominant role. This is reasonable since there is only ductile flow when the cut depth is less than 1 µm [18]. For this study, the cut depth (from Equation (10)) is 0.5 µm to 4 µm, so there is lateral cracking and plastic flow. Figure 9 shows the relationship between the working load and the ratio of brittle/ductile removal rates. Increasing the working load increases the cut depth so there is more evidence of lateral brittle fracture and the kerf width increases. For wire-saw machining, the MRR is increased when the working load (*P*) is increased. If the working load exceeds a threshold, the steel wire can break easily. The cutting path can also become crooked. Therefore, the working load must not exceed a specified threshold value. If the wire speed (*S*) is increased, the amount of grains entering the machining zone from the slurry within a specific time increases, so significantly more material is removed.

From Equation (20), the larger the grit size A, the greater is the machining efficiency. However, the surface roughness of the wafer also increases. Suwabe et al. [24] showed the addition of smaller grains in the slurry results in a smoother wafer. However, when all of the particles in the slurry are smaller than #1000, a further increase in the working load or the wire speed produces no increase in the volume that is removed. Figure 10 shows the theoretical values for material removed for abrasives of mesh size #600 (solid line) and size #1000 (dash line). There is no difference in the amount of material removed when the grit is too small. The higher the material hardness, the greater the normal cutting force ($F_{Y}$), so the MRR is decreased. Higher material hardness results in brittle fracture of the material. The coefficient of friction has an effect on crack propagation [25]. Figure 3b shows when the coefficient of friction decreases, a lateral crack propagates almost parallel to the specimen surface. This results in brittle fracture and more material being removed. This information is confirmed by Equation (20).

## 4. Conclusions

Slicing ceramic technology using WSM still has disadvantages in terms of the MRR deviation. However, WSM increases productivity and the MRR for slicing ceramics, which decreases the manufacturing cost. This paper developed a mathematical model for SWSM of brittle material to study the effect of important parameters (wire speed and working load) on material removal. Actual SWSM experiments on MRR are conducted to verify the results from the mathematical model study. The theoretical results agree with experimental data and practical experience. The following conclusions were drawn:

Increasing the working load will decrease MRR from the machining zone for SWSM because the decreasing clearance between the wire and the surface of the workpiece prevents the abrasive grains from entering the working area.

The lower the wire speed is, the higher the $N_{Y}$ value. Therefore, the faster the wire speed, the greater the value of n_1_, and the smaller the volume removed by a single abrasive (*V*).

A small amount of material is removed by plastic deformation, and brittle fracture plays the dominant role in SWSM ceramics.

Increasing the working load increases the cut depth, so there is more evidence of lateral brittle fracture and the kerf width increases.

The higher the material hardness, the greater the normal cutting force ($F_{Y}$), so the MRR is decreased. However, higher material hardness results in brittle fracture of the material in SWSM ceramics.

## Figures and Tables

**Figure 1 materials-13-04324-f001:**
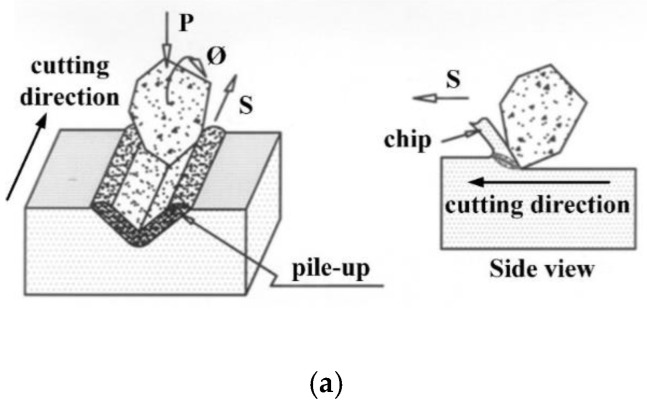
Illustration of wire-sawing: (**a**) fine chips removed by plastic deformation; (**b**) debris generated by brittle fracture.

**Figure 2 materials-13-04324-f002:**
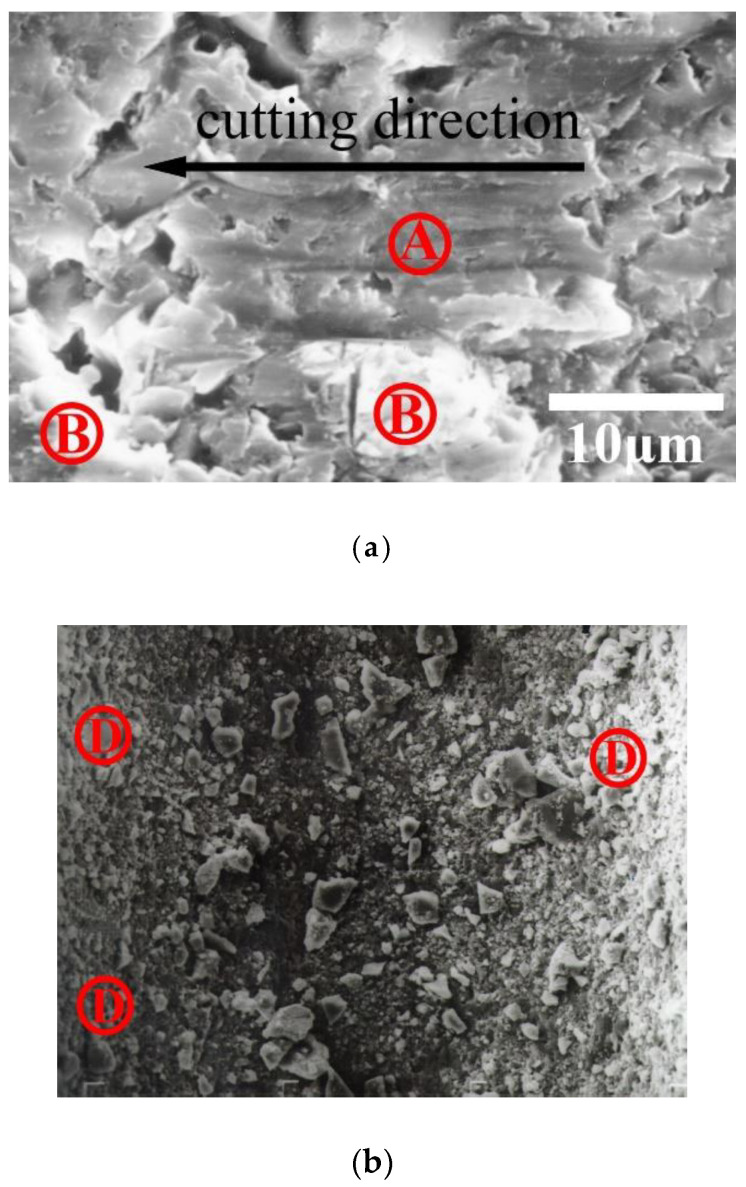
Wire-saw machining of Al_2_O_3_ using SiC (working load of 1.27 N, wire speed of 2.8 m/s): (**a**) plastic deformation at position A and brittle fracture at position B; (**b**) debris at position D; (**c**) energy-dispersive spectrometry results for the debris shown in Figure 2b.

**Figure 3 materials-13-04324-f003:**
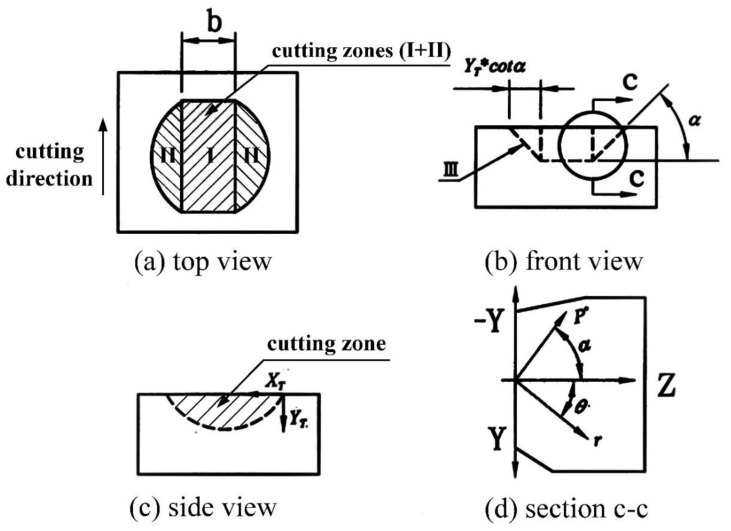
Schematic diagram of the volume removed by a single abrasive particle (Zone I: plastic deformation region, Zone II: brittle fracture region, Zone III: predicted lateral fracture path, b: cutting width of the abrasive particle, P^*^: minimum threshold load for the lateral crack, tanα = µ).

**Figure 4 materials-13-04324-f004:**
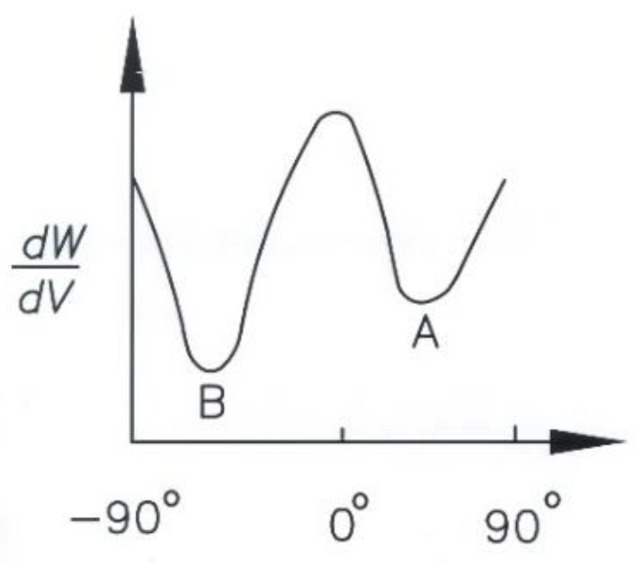
Illustration of fracture initiation.

**Figure 5 materials-13-04324-f005:**
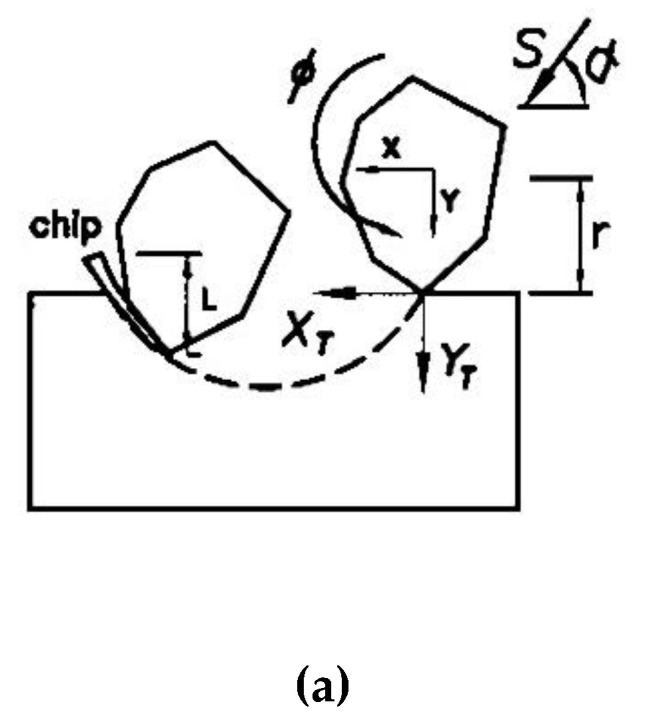
Schematic diagram of a two-dimensional cutting process using an abrasive particle: (**a**) inertial force only; (**b**) working load (P); and wire speed (S).

**Figure 6 materials-13-04324-f006:**
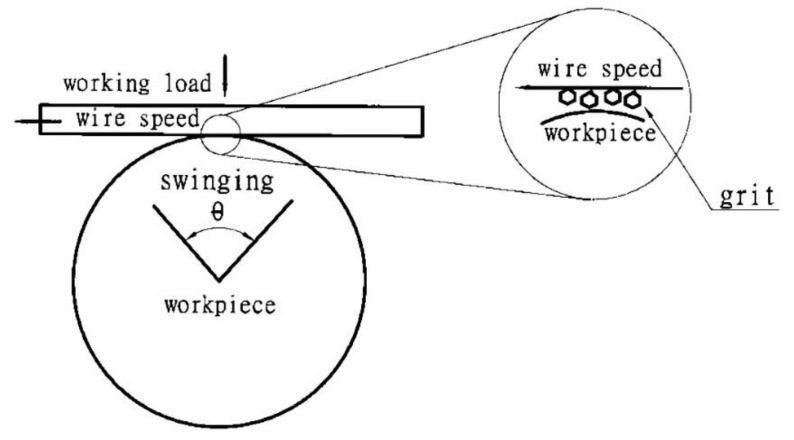
Model of a swinging wire-sawing.

**Figure 7 materials-13-04324-f007:**
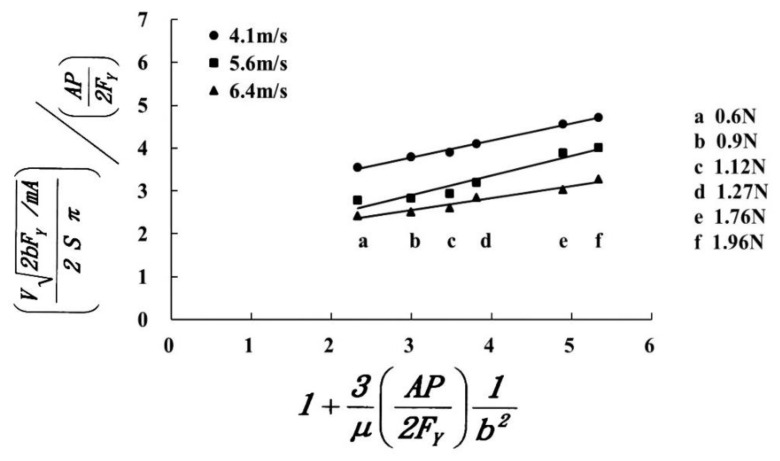
Experimental results for the volume of Al_2_O_3_ that is removed.

**Figure 8 materials-13-04324-f008:**
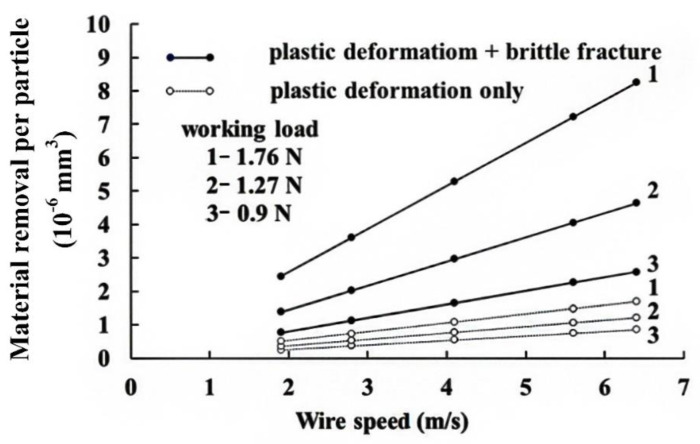
Theoretical value for material removed by a single grit.

**Figure 9 materials-13-04324-f009:**
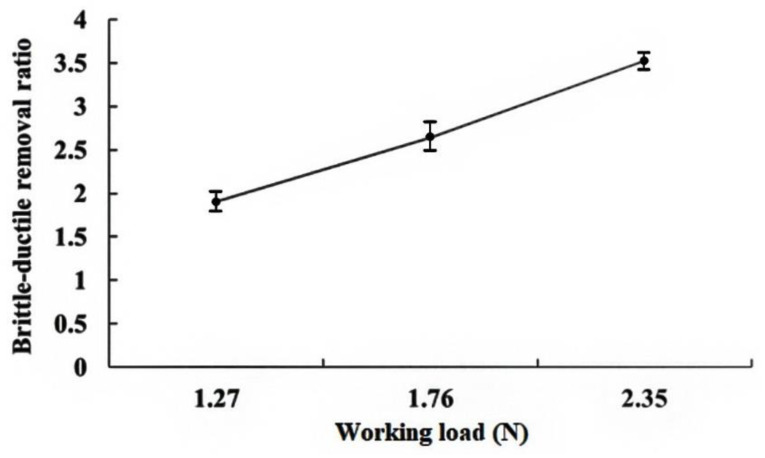
Ratio of brittle fracture to ductile removal rate.

**Figure 10 materials-13-04324-f010:**
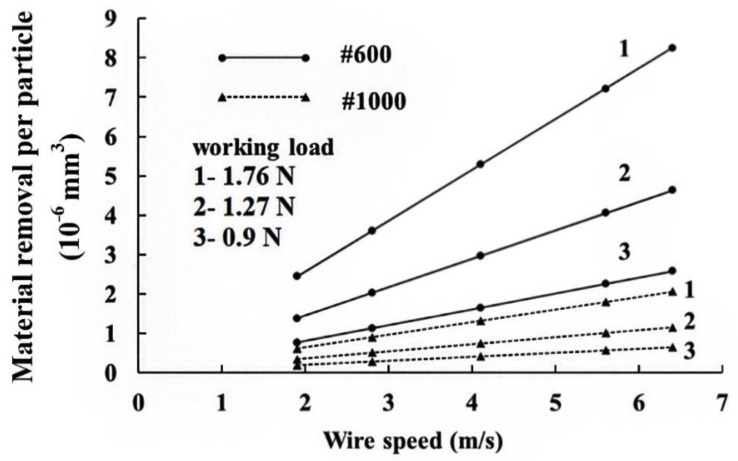
Material removed by different size of grit.

**Table 1 materials-13-04324-t001:** Experimental conditions.

Item	Specification
Workpiece (Diameter)	Al_2_O_3_ (ϕ8 mm)
Slurry contents	SiC + Water
Grains (Diameter)	GC# 600 (27 µm)
Concentration (wt.%)	10
Wire diameter (mm)	ϕ0.24 ± 0.05 (Stainless wire)
Wire tension (N)	18
Wire speed (m/s)	4.1, 5.6 and 6.4
Working load (N)	0.60, 0.90, 1.12, 1.27, 1.76 and 1.96
Frequency (Hz)	0.8
Vibration angle (θ)	60°

**Table 2 materials-13-04324-t002:** Experimental conditions and results.

	*P* (N)	*S* (m/s)	*V*	*N_X_*	*N_Y_*
a	0.6	4.1	1.31 × 10^−7^	2.325	3.550
	0.6	5.6	1.41 × 10^−7^	2.325	2.791
	0.6	6.4	1.10 × 10^−7^	2.325	2.428
b	0.9	4.1	2.04 × 10^−7^	2.990	3.680
	0.9	5.6	2.08 × 10^−7^	2.990	2.750
	0.9	6.4	2.14 × 10^−7^	2.990	2.480
c	1.12	4.1	2.69 × 10^−7^	3.474	3.900
	1.12	5.6	2.76 × 10^−7^	3.474	2.940
	1.12	6.4	2.72 × 10^−7^	3.474	2.610
d	1.27	4.1	3.20 × 10^−7^	3.805	4.100
	1.27	5.6	3.40 × 10^−7^	3.805	3.197
	1.27	6.4	3.47 × 10^−7^	3.805	2.850
e	1.76	4.1	5.00 × 10^−7^	4.887	4.570
	1.76	5.6	5.10 × 10^−7^	4.887	3.880
	1.76	6.4	5.10 × 10^−7^	4.887	3.040
f	1.96	4.1	5.70 × 10^−7^	5.330	4.710
	1.96	5.6	6.30 × 10^−7^	5.330	4.010
	1.96	6.4	6.25 × 10^−7^	5.330	3.270

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
