# Peer review of "Slicing Ceramics on Material Removed by a Single Abrasive Particle"

_materials, 2020, doi:10.3390/ma13194324_

Round 1

Reviewer 1 Report

Review of materials-921121-peer-review-v1: “Theoretical derivation and a single abrasive particle on material removed of slicing ceramic”

The subject of the paper is relevant with the topics of the journal.

I would suggest the authors to incorporate the following in order to improve the quality of the paper:

  • In line 34, explain abbreviations SWSM and MWSM
  • For Fig. 1, 2, 4, 5, 6 images, check for copyright issues.
  • Conclusions should expand more the findings and the authors’ contribution.

My proposal to the editor is to accept the paper after minor revisions.

Author Response

Reviewer #1

The subject of the paper is relevant with the topics of the journal.

I would suggest the authors to incorporate the following in order to improve the quality of the paper:

  1. In line 34, explain abbreviations SWSM and MWSM

Response: It is revised according to the reviewer’s suggestion. MWSM means wire saw machining with multi-wires in wire-sawing. SWSM means wire saw machining with single-wire in wire-sawing.

  1. For Fig. 1, 2, 4, 5, 6 images, check for copyright issues.

Response: Figs. 1, 2, 4, 5 and 6 are original, so it is not copyright problems.

  1. Conclusions should expand more the findings and the authors’ contribution.

Response: It is revised according to the reviewer’s suggestion.

Reviewer 2 Report

The present work deals with the study of wire saw machining on ceramic material. In particular, the authors derived a mathematical model to calculate the material removed by the sawing process, based on the equation of motion of particles. Then, various experimental tests were carried out to validate the model and to study the effect of different parameter sets on the removal of hard and brittle materials. The topic of the work is interesting and the paper is well organized; however, the article should be revised before publication, according to the following suggestions:

  1. There are some language errors and typos that should be fixed.
  2. The formulas at lines 263 and 284 should be checked.
  3. The values of the working load and the wire speed of table 2 do not agree with those reported in table 1.
  4. The number of runs and the obtained standard deviation should be reported for each parameter set of the experimental campaign. Moreover, the correspondence between the numerical and experimental results should also be shown on the graph.
  5. Wire cutting has been the subject of studies and research for years by many researchers, but the bibliography, presented by the authors, appears quite poor. Relevant contributions are lacking. See, for instance, “Turchetta, S., Sorrentino, L., Bellini, C. A method to optimize the diamond wire cutting process (2017) Diamond and Related Materials, 71, pp. 90-97. DOI: 10.1016/j.diamond.2016.11.016” and “Yilmazkaya, E., Ozcelik, Y. The Effects of Operational Parameters on a Mono-wire Cutting System: Efficiency in Marble Processing (2016) Rock Mechanics and Rock Engineering, 49 (2), pp. 523-539. DOI: 10.1007/s00603-015-0743-9”.

Author Response

Reviewer #2

The present work deals with the study of wire saw machining on ceramic material. In particular, the authors derived a mathematical model to calculate the material removed by the sawing process, based on the equation of motion of particles. Then, various experimental tests were carried out to validate the model and to study the effect of different parameter sets on the removal of hard and brittle materials. The topic of the work is interesting and the paper is well organized; however, the article should be revised before publication, according to the following suggestions:

  1. There are some language errors and typos that should be fixed.

Response: It is revised according to the reviewer’s suggestion.

  1. The formulas at lines 263 and 284 should be checked.

Response: The formulas at lines 263 and 284 have been checked according to the reviewer’s suggestion.

  1. The values of the working load and the wire speed of table 2 do not agree with those reported in table 1.

Response: It is revised according to the reviewer’s suggestion.

  1. The number of runs and the obtained standard deviation should be reported for each parameter set of the experimental campaign. Moreover, the correspondence between the numerical and experimental results should also be shown on the graph.

Response: Each experiment was replicated three times to verify the theoretical model developed in this study. It is added according to the reviewer’s suggestion.

  1. Wire cutting has been the subject of studies and research for years by many researchers, but the bibliography, presented by the authors, appears quite poor. Relevant contributions are lacking. See, for instance, “Turchetta, S., Sorrentino, L., Bellini, C. A method to optimize the diamond wire cutting process (2017) Diamond and Related Materials, 71, pp. 90-97. DOI: 10.1016/j.diamond.2016.11.016” and “Yilmazkaya, E., Ozcelik, Y. The Effects of Operational Parameters on a Mono-wire Cutting System: Efficiency in Marble Processing (2016) Rock Mechanics and Rock Engineering, 49 (2), pp. 523-539. DOI: 10.1007/s00603-015-0743-9”.

Response: It is revised according to the reviewer’s suggestion.

Reviewer 3 Report

Authors present an important topic related to processing ceramic materials. 

My comments are as follows:

  1. The title should be corrected: it is really difficult to understand the real idea of the paper when reading the title - on material removed by a single abrasive particle? slicing ceramics or ceramic materials.
  2. Figures are not proportional. The sizes of descriptions are different, not uniform. Captions of a), b) etc. should be given in the main caption of a figure. 
  3. Descriptions of the symbols should be given as the symbols are shown in the text for the first time, e.g., Fig. 1 and Fig. 5 - P and S.
  4. Figure 7: The names of axis should be given with units, not equations.
  5. Fig. 8 Material removal not removel, speed not speem. It could be better " material removal per particle [10-6 mm]
  6. Fig. 9: units are missing
  7. Fig. 8,10: Can't write "1=1.76N". Should be 1- 1.76 N, etc. 
  8. Conclusions are very week and general. Before the conclusions, the discussion on the comparison of models developed by Authors and presented in other papers should be given and the differences should be emphasized. 

Author Response

Reviewer #3

Authors present an important topic related to processing ceramic materials. 

My comments are as follows:

  1. The title should be corrected: it is really difficult to understand the real idea of the paper when reading the title - on material removed by a single abrasive particle? slicing ceramics or ceramic materials.

Response: It is revised according to the reviewer’s suggestion.

  1. Figures are not proportional. The sizes of descriptions are different, not uniform. Captions of a), b) etc. should be given in the main caption of a figure.

Response: It is revised according to the reviewer’s suggestion.

  1. Descriptions of the symbols should be given as the symbols are shown in the text for the first time, e.g., Fig. 1 and Fig. 5 - P and S.

Response: The explanation of all symbols has shown in manuscript (Nomenclature).

  1. Figure 7: The names of axis should be given with units, not equations.

Response: In fact, the unit of axis in Fig. 7 is is expressed in non-dimensional form.

  1. 8 Material removal not removel, speed not speem. It could be better " material removal per particle [10-6 mm3]

Response: It is revised according to the reviewer’s suggestion.

  1. 9: units are missing

Response: It is revised according to the reviewer’s suggestion.

  1. 8,10: Can't write "1=1.76N". Should be 1- 1.76 N, etc.

Response: It is revised according to the reviewer’s suggestion.

  1. Conclusions are very week and general. Before the conclusions, the discussion on the comparison of models developed by Authors and presented in other papers should be given and the differences should be emphasized.

Response: It is revised according to the reviewer’s suggestion.

Round 2

Reviewer 3 Report

In can be accepted in the presented form, although it must meet the journal requirements regarding formatting.